# Evaluating Uncertainty Quantification approaches for Neural PDEs in scientific applications

**Vardhan Dongre**
University of Illinois Urbana-Champaign
vdongre2@illinois.edu

**Gurpreet Singh Hora**
Columbia University
gh2546@columbia.edu

## Abstract

The accessibility of spatially distributed data, enabled by affordable sensors, field, and numerical experiments, has facilitated the development of data-driven solutions for scientific problems, including climate change, weather prediction, and urban planning. Neural Partial Differential Equations (Neural PDEs), which combine deep learning (DL) techniques with domain expertise (e.g., governing equations) for parameterization, have proven to be effective in capturing valuable correlations within spatiotemporal datasets. However, sparse and noisy measurements coupled with modeling approximation introduce aleatoric and epistemic uncertainties. Therefore, quantifying uncertainties propagated from model inputs to outputs remains a challenge and an essential goal for establishing the trustworthiness of Neural PDEs. This work evaluates various Uncertainty Quantification (UQ) approaches for both Forward and Inverse Problems in scientific applications. Specifically, we investigate the effectiveness of Bayesian methods, such as Hamiltonian Monte Carlo (HMC) and Monte-Carlo Dropout (MCD), and a more conventional approach, Deep Ensembles (DE). To illustrate their performance, we take two canonical PDEs: Burger's equation and the Navier-Stokes equation. Our results indicate that Neural PDEs can effectively reconstruct flow systems and predict the associated unknown parameters. However, it is noteworthy that the results derived from Bayesian methods, based on our observations, tend to display a higher degree of certainty in their predictions as compared to those obtained using the DE. This elevated certainty in predictions suggests that Bayesian techniques might underestimate the true underlying uncertainty, thereby appearing more confident in their predictions than the DE approach.

## 1 Introduction

While conventional Deep Learning-based approaches provide promising avenues for the scientific domain, they often struggle to uphold the physical realizability of the solutions. Physics-Informed Neural Networks (PINNs), as introduced by [11], excel at incorporating soft physical constraints within the neural network optimization process, leading to better outcomes. However, due to noisy and limited data, the accuracy of these models can degrade significantly. Adopting these methods, in principle, requires the models and their predictions to be reliable, due to which our ability to quantify the uncertainties involved in the process becomes significantly valuable.

The UQ problem has two intimately coupled components [8]. The first pertains to the forward propagation of uncertainty from model parameters to model outputs, and the second component involves the estimation of the parametric uncertainties themselves based on available data. The focus of this work is to quantify the total uncertainty from both components. The task of UQ in scientific machine learning is complex, given the stochastic processes in science, model overparameterization, and data noise [e.g., 5, 3, 2, 17]. Past research has employed diverse Bayesian and Deterministic

strategies to quantify model uncertainty, but a comparative understanding of these methods' efficacy remains elusive. This study seeks to fill this gap by systematically comparing various uncertainty quantification techniques, including Hamiltonian Monte Carlo (HMC), Monte Carlo Dropouts (MCD), and Deep Ensembles (DE). We apply these methods to forward and inverse problems in two canonical PDEs - the Burger's equation and the Navier-Stokes equation, illustrating their performance in reconstructing flow systems and predicting parameters from sparse noisy measurements.

## 2 Forward Problems

Consider a parameterized and non-linear PDE that characterizes the behavior of a physical system, defined as

$$\mathcal{L}_{\mathbf{x}}[\mathbf{u}; \lambda] = \mathbf{f}(\mathbf{x}, t), \mathbf{x} \in \Omega, t \in [0, T], \tag{1}$$

where $\mathbf{u}(\mathbf{x}, t)$ denotes the latent state (aka solution field), the $\mathcal{L}_{\mathbf{x}}[.; \lambda]$ is a general differential operator parameterized by $\lambda$, $\mathbf{f}(\mathbf{x}, t)$ is the forcing term which refers to any external influences on the system, while $\Omega \subset \mathbb{R}^D$ is the bounded domain in a d-dimensional physical space.

Given this framework and noisy measurements of $\mathbf{u}(\mathbf{x}, t), \mathbf{f}(\mathbf{x}, t)$, the goal is to infer the latent state $\mathbf{u}(\mathbf{x}, t)$ of the dynamical system. In forward problems, PINNs as well as their Bayesian variants B-PINNs are typically used as surrogates $\widetilde{\mathbf{u}}(\mathbf{x}, t; \theta)$, to infer either point estimates or posterior distributions of this latent state. In the Bayesian framework, the parameters $\theta$ of the surrogates have a prior distribution $P(\theta)$ and its formulation is defined as:

$$\widetilde{\mathbf{f}}(\mathbf{x}, t; \theta) := \mathcal{L}_{\mathbf{x}}[\widetilde{\mathbf{u}}(\mathbf{x}, t; \theta); \lambda] \tag{2}$$

$P(\mathcal{D}|\theta)$ represents the likelihood while the Bayes' Theorem estimates the final posterior distribution.

$$p(\theta|\mathcal{D}) = \frac{P(\mathcal{D}|\theta)P(\theta)}{P(\mathcal{D})} \cong P(\mathcal{D}|\theta)P(\theta) \tag{3}$$

To approximate the posterior distribution, we employ both Bayesian methods like HMC and MCD as well as deterministic DE approach. HMC is an efficient Markov Chain Monte Carlo (MCMC) sampling method that uses concepts from Hamiltonian Dynamics and utilizes momentum variables to guide the proposals in the Markov chain, which can lead to faster convergence and better exploration of the target distribution. Given the continuous nature of Hamiltonian dynamics, leapfrog integration is used as a numerical technique to discretize and update the momentum and position variables in a staggered manner over discrete time steps. In our Bayesian methodology, we posit an independent Gaussian distribution as the prior $P(\theta)$. For HMC, parameters for Burger's (Navier-Stokes) equation include a leapfrog step of 50 (50), an initial time step of 0.1 (0.01), 1000 (5000) burn-in steps, and a sampling size of 100 (200). With DE, we assemble an ensemble of PINNs equivalent in number to the HMC samples, set at 100 (200). For MCD, we induce variance by sporadically dropping neurons at a 1% (1%) dropout rate during each training iteration. To gauge prediction uncertainty, we execute 100 (200) inferences with HMC. For DE, we acquire 100 (200) predictions from each ensemble member, and for MCD, we undertake forward network propagation 100 (200) times, maintaining the established dropout rate.

### 2.1 1-D Burger's Equation

Burger's equation is a PDE that arises in fluid dynamics and represents a combination of diffusion and convection processes. It has wide applications in various scientific domains, including traffic flow modeling [7], acoustics and sound propagation [9], and material transport in porous media [13]. For this work, we consider a one-dimensional Burger's equation with Dirichlet boundary condition and sinusoidal initial conditions:

$$\frac{\partial u}{\partial t} + u \cdot \nabla u - \frac{0.01}{\pi} \nabla^2 u = 0, \ \ x \in [-1, 1], \ t \in [0, 1], \tag{4}$$
$$u(0, x) = -sin(\pi x),$$
$$u(t, -1) = u(t, 1) = 0,$$

where $x$ represents the spatial location, $t$ represents time, $u(x, t)$ represents the velocity of the fluid, and $\nabla$ and $\nabla^2$ represents gradient and Laplacian operators.

To find its exact solution, we employ the Chebfun package [12], utilizing spectral Fourier discretization with 512 modes and a fourth-order explicit Runge-Kutta temporal integrator featuring a time step of $\Delta t = 10^{-6}$. For a more comprehensive understanding, consult the methodology detailed in [11]. Here, we operate under the assumption that the exact solution remains unknown. Instead, we rely on noisy sensors that capture 2000 spatiotemporal readings for $u$ and $f$. The noise in these measurements adheres to a Gaussian distribution with scales $\epsilon_f \sim \mathcal{N}(0, 0.1^2)$ and $\epsilon_u \sim \mathcal{N}(0, 0.1^2)$. In our experiments, we employ a multilayer perceptron (MLP) neural network consisting of eight hidden layers, each comprising 20 neurons with tanh non-linearity.

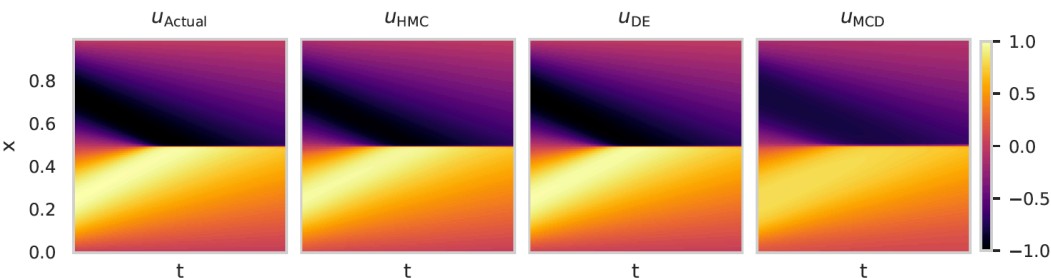

Figure 1: One-dimensional Burgers equation - forward problem: comparison of the spatiotemporal evolution of predictive mean and exact solutions for $u$. HMC represents Hamiltonian Monte Carlo, DE represents Deep Ensembles, MCD represents Monte Carlo Dropout, and Actual represents the exact solution.

Figure 1 presents a comparative analysis of predictive spatiotemporal means derived from three distinct approaches, juxtaposed against the reference actual solution, denoted as ($u_{\text{Actual}}$). An initial visual assessment of these predictions reveals their impressive fidelity to the exact solutions, effectively reconstructing the solution to the Burgers equation in both space and time from sparse measurements. Notably, within Figure 1, it becomes evident that both the HMC ($u_{\text{HMC}}$) and DE ($u_{\text{DE}}$) approaches provide accurate predictions of the magnitude of $u$, closely matching the actual solution. In contrast, the MCD ($u_{\text{MCD}}$) consistently underestimates the magnitude of the solution.

The spatiotemporal color maps of the predictive mean do not effectively convey the model's confidence in its predictive capabilities. To investigate the uncertainties in the predictions, the predictive means along with the corresponding two standard deviation confidence intervals generated by three different methods, HMC, MCD, and DE, for the variable $u$ at three distinct time snapshots, namely, $t = 0.50s, 0.75s, 0.90s$ are illustrated in figure 2. From visual inspection, it is evident that both the HMC and DE approaches provide reasonably accurate posterior estimations of the variable $u$ at all three time-snapshots. Moreover, the error between these predictive means and the actual solution remains predominantly within the bounds of the two standard deviations. In contrast, the MCD approach exhibits discrepancies from the actual solution across all temporal snapshots, although these discrepancies tend to diminish as time progresses. It is noteworthy that, for $t = 0.50s$, a significant portion of the error falls outside the two standard deviation confidence intervals. However, as time advances, the performance of the MCD approach noticeably improves. It is also important to highlight that all three approaches effectively capture the formation of shocks, a challenging task even for classical numerical methods.

Our results show that the Bayesian approaches, i.e., HMC render overconfident results while model outputs from DE and MCD are appropriately conservative as expected [2]. In conclusion, both HMC and DE seem to be superior in terms of both prediction accuracy and uncertainty quantification, especially when compared to the results from MCD. While MCD's performance improves with time, its initial underestimation of uncertainty could be problematic, especially in scenarios where early-stage predictions are critical.

## 2.2 2-D Navier Stokes Equation

Our next example delves into a practical scenario involving the flow of an incompressible fluid, a phenomenon elegantly described by the renowned Navier-Stokes equations. These equations stand

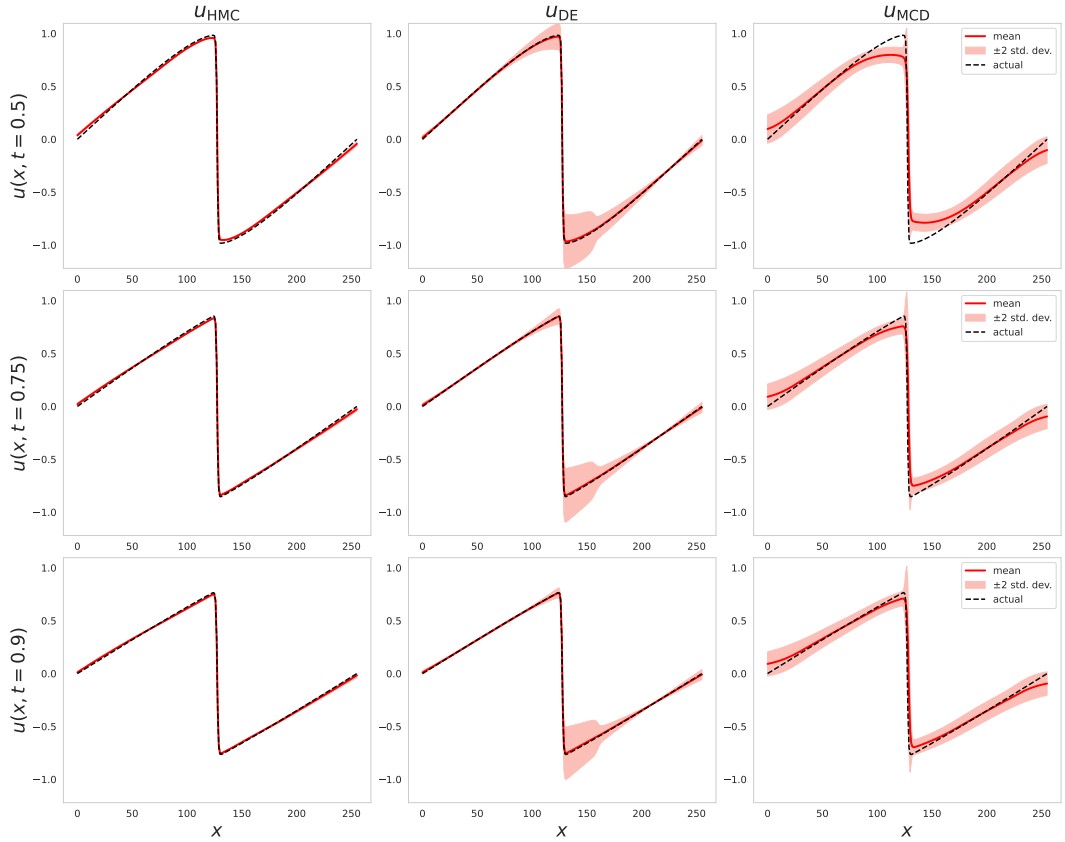

Figure 2: One-dimensional Burgers equation - forward problem: comparison of the predicted and exact solutions corresponding to the three temporal snapshots denoted by $t \in \{0.50s, 0.75s, 0.90s\}$ from different methods.

as a cornerstone in the realm of scientific and engineering dynamics, offering profound insights and applications. They find utility in several geophysical and engineering domains, such as climate prediction [10], air pollution [1], aerodynamics of aircraft and cars [4, 6, 15], and blood circulation [14, 16]. In this work, we consider a prototype problem of incompressible flow past a cylinder, and the governing equation can be defined as follows:

$$\frac{\partial \mathbf{u}}{\partial t} + \lambda_1 \mathbf{u} \cdot \nabla \mathbf{u} + \nabla \mathbf{u} - \lambda_2 \nabla^2 u = 0, \tag{5}$$

$$\nabla \cdot \mathbf{u} = 0,$$

where $\mathbf{u} = \{u, v\}$ represents the velocity in $x$ and $y$ direction, $p$ represents the pressure of the fluid, $t$ represent time, and $\lambda = \{\lambda_1, \lambda_2\}$ are the parameters and for the forward problems $\lambda_1$ is set to 1 and $\lambda_2$ to $10^{-2}$. Given the multidimensional nature of this problem, it offers a challenging testbed for the Bayesian approach to quantify uncertainties in both the velocity and pressure fields. It is important to emphasize that pressure measurements are not included in the model training; instead, the neural network predicts them based on the governing equation. To generate the exact solutions, we leverage the data provided for the work by [11], and readers are advised to refer to the same for more details.

Similarly, we operate under the assumption that the precise solution remains elusive while our sensors diligently capture 5000 spatiotemporal readings for both $u$ and $f$. These measurements exhibit a Gaussian distribution with scales $\epsilon_f \sim \mathcal{N}(0, 0.1^2)$ and $\epsilon_u \sim \mathcal{N}(0, 0.1^2)$. To effectively approximate the latent variables in the Navier-Stokes equation—namely, $u$, $v$, and $p$ – we employ an MLP network comprising ten hidden layers, each housing 20 neurons with a tanh non-linearity.

The predictive means obtained from HMC, DE, and MCD for the velocity components $u$ and $v$ are compared with the reference actual solution, figure 3. It is readily apparent from the figure

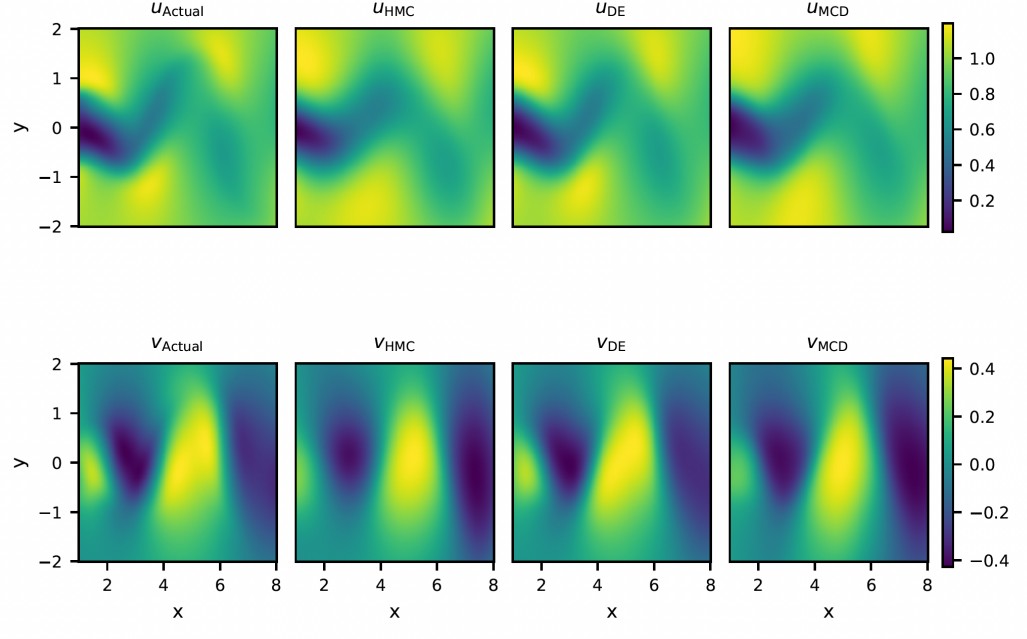

Figure 3: Navier-Stokes equation - forward problem: Instantaneous predictive mean for velocity component in $x$-direction $u$ (top) and $y$-direction $v$ (bottom) using HMC, DE, and MCD approaches is compared against the actual instantaneous $u$ and $v$ components at a representative time instant.

that all three models have adeptly reconstructed the $u$ and $v$ velocity components. This remarkable accuracy is achieved despite the challenges posed by the noisy, scattered sensor data across the entire spatiotemporal domain. Moreover, we examine the $L_1$ norm-based error between the actual and predictive mean values to gain further insights, as presented in figure 4a. Notably, the DE approach exhibits the closest agreement with the actual solutions. In contrast, the error for the HMC and MCD approaches is roughly three times higher than that observed with the DE approach for both velocity components. Importantly, it is worth noting that, across all proposed methodologies, the errors consistently remain within the bounds of two standard deviations, as illustrated in figure 4b. This observation underscores our confidence in the predictions generated by these various approaches, as they remain well within the established confidence interval.

## 3 Inverse Problems

Inverse problems involve determining a system's underlying parameters $\lambda$ and physical properties from observable data.

In the context of our study, B-PINN offers a systematic approach to tackle inverse problems. We can quantify uncertainties in the estimated parameters by propagating uncertainties through the network and leveraging the Bayesian framework. Similar to the framework described in equations [2-3], apart from a surrogate for $\theta$, we also assign a prior distribution for $\lambda$, which can be independent of the prior for $\theta$. The likelihood is then defined as $P(\mathcal{D}|\theta, \lambda)$, and we then calculate the joint posterior of $[\theta, \lambda]$:

$$p(\theta, \lambda | \mathcal{D}) = \frac{P(\mathcal{D}|\theta, \lambda) P(\theta, \lambda)}{P(\mathcal{D})} \cong P(\mathcal{D}|\theta, \lambda) P(\theta, \lambda) = P(\mathcal{D}|\theta, \lambda) P(\theta) P(\lambda) \quad (6)$$

The PDE considered for the inverse problem is the same Navier-Stokes equation (Refer equation 5). However, in the context of the inverse problems, the parameters $\lambda = \{\lambda_1, \lambda_2\}$ are now considered unknown. The primary objective here is to identify the values of unknown parameters based on the limited measurements of $f$ and $\mathbf{u} = [u, v]$ outlined in section 2.2.

The MLP model we employ for the inverse problem has ten hidden layers with 40 neurons in each layer and tanh non-linearity. The predicted values of $\lambda$ from considered approaches are displayed

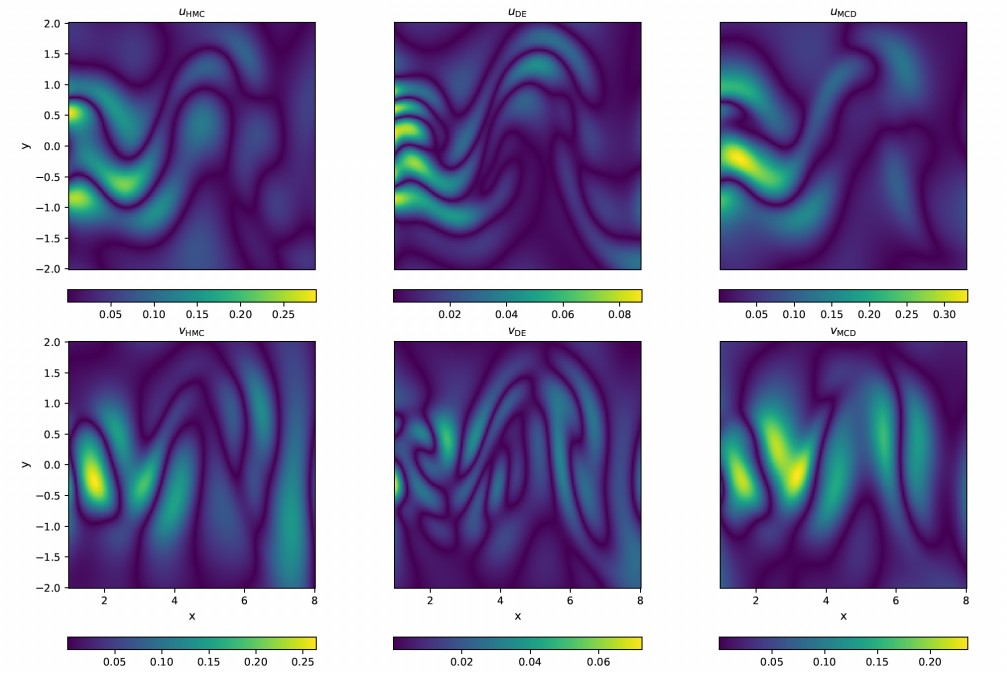

(a) Predictive errors for velocity component in $x$-direction $u$ (top) and $y$-direction $v$ (bottom) from different methods at a representative time instant.

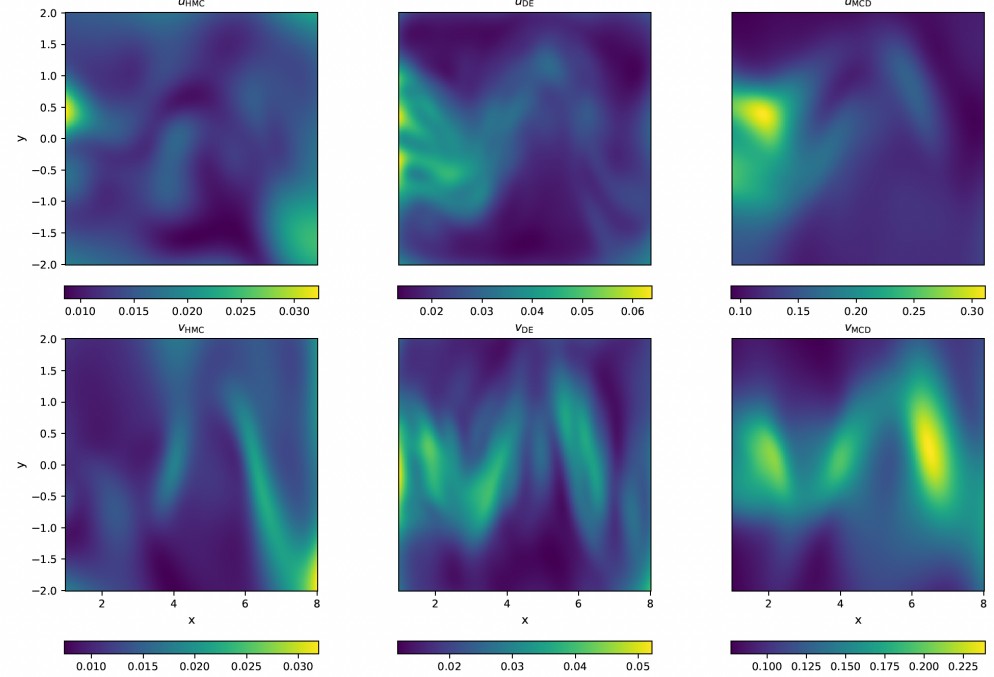

(b) Two standard deviations $(\sigma)$ for velocity component in $x$-direction $u$ (top) and $y$-direction $v$ (bottom) from different methods at a representative time instant.

Figure 4: Navier-Stokes equation - forward problem.

| | HMC | DE | MCD |
|---|---|---|---|
| $\lambda_1$ (mean $\pm$ std) | $0.758 \pm 0.0$ | $0.957 \pm 0.024$ | $0.843 \pm 0.075$ |
| $\lambda_2$ (mean $\pm$ std) | $0.017 \pm 2.13\mathrm{e}{-06}$ | $0.014 \pm 0.001$ | $0.015 \pm 0.058$ |

Table 1: Navier Stokes equation - inverse problem : Predictions for $\lambda_1, \lambda_2$ using HMC, DE, MCD; actual values for $\lambda_1 = 1.0, \lambda_2 = 0.01$

in Table 1. The DE method has provided relatively precise estimates, reflecting a good degree of certainty in its predictions. This suggests that ensemble techniques effectively capture these parameters' underlying distributions. While HMC provides high confidence in its estimates, the absence of uncertainty is unrealistic, and this overconfidence could be a sign of the model not capturing all sources of uncertainty. MCD provides a broader uncertainty estimation, which might be capturing more sources of uncertainties, but it could also be overestimating the uncertainty in the parameters. The wider confidence intervals for MCD could either mean that MCD is being more cautious or it's not as effective in pinpointing the true parameter values. These findings underscore the effectiveness of the DE approach in not only identifying the unknown parameters but also quantifying the uncertainty arising from the sparse and noisy sensor measurements.

## 4 Summary

This research comparatively evaluates various UQ approaches, with particular emphasis on Bayesian methods and the Deep Ensemble (DE) technique. While all approaches, including DE, HMC, MCD, effectively reconstruct flow systems for the two examples considered, Bayesian methods demonstrate higher certainty in predictions but may underestimate the total uncertainty, thereby appearing overly confident. In contrast, while offering more conservative certainty estimates, the DE method is computationally more demanding. We also acknowledge that the performance of these methods improves by inferring more samples, but due to computational constraints, we restrict the numbers to 100 (200) for Burger's (Navier Stokes) Equation. The study underscores the need for balancing predictive certainty, computational efficiency, and accuracy when using Bayesian or DE approaches for flow system modeling and parameter prediction.

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
