# OpenReview forum: "Evaluating Uncertainty Quantification approaches for Neural PDEs in scientific applications"
_NeurIPS.cc/2023/Workshop/AI4Science — NeurIPS2023-AI4Science Poster_

### Official Review · Reviewer_QxCp · 2023-10-25
**Comparison of various Uncertainty Quantification methods  on Neural Partial Differential Equations**

**Rating:** 7
**Confidence:** 3

**Review:**

Pros
Well written
Heat map representation of model uncertainties looks interesting

---

### Meta-Review · Area_Chair_wpzk · 2023-10-27

**Recommendation:** Accept (Poster)
**Confidence:** 3

**Metareview:**

This paper benchmark different Uncertainty Quantification (UQ) approaches. Those methods include Hamiltonian Monte Carlo (HMC) and Monte-Carlo Dropout (MCD), and Deep Ensembles (DE). The problem that are exploited are Burger's equation and the Navier-Stokes equation. The results can provide valuable information for the community. Hence, I recommend the acceptance of the paper as a poster.